# Compressing Images by Encoding Their Latent Representations with Relative Entropy Coding

**Gergely Flamich**[*]
Department of Engineering
University of Cambridge
gf332@cam.ac.uk

**Marton Havasi**[*]
Department of Engineering
University of Cambridge
mh740@cam.ac.uk

**José Miguel Hernández-Lobato**
Department of Engineering
University of Cambridge,
Microsoft Research,
Alan Turing Institute
jmh233@cam.ac.uk

## Abstract

Variational Autoencoders (VAEs) have seen widespread use in learned image compression. They are used to learn expressive latent representations on which downstream compression methods can operate with high efficiency. Recently proposed 'bits-back' methods can indirectly encode the latent representation of images with codelength close to the relative entropy between the latent posterior and the prior. However, due to the underlying algorithm, these methods can only be used for lossless compression, and they only achieve their nominal efficiency when compressing multiple images simultaneously; they are inefficient for compressing single images. As an alternative, we propose a novel method, Relative Entropy Coding (REC), that can directly encode the latent representation with codelength close to the relative entropy for single images, supported by our empirical results obtained on the Cifar10, ImageNet32 and Kodak datasets. Moreover, unlike previous bits-back methods, REC is immediately applicable to lossy compression, where it is competitive with the state-of-the-art on the Kodak dataset.

## 1 Introduction

The recent development of powerful generative models, such as Variational Autoencoders (VAEs) has caused a great deal of interest in their application to image compression, notably Ballé et al. (2016a, 2018); Townsend et al. (2020); Minnen & Singh (2020). The benefit of using these models as opposed to hand-crafted methods is that they can adapt to the statistics of their inputs more effectively, and hence allow significant gains in compression rate. A second advantage is their easier adaptability to new media formats, such as light-field cameras, 360° images, Virtual Reality (VR), video streaming, etc. for which classical methods are not currently applicable, or are not performant.

VAEs consist of two neural networks, the encoder and the decoder. The former maps images to their latent representations and the latter maps them back. Compression methods can operate very efficiently in latent space, thus realizing a non-linear transform coding method (Goyal, 2001; Ballé et al., 2016b). The sender can use the encoder to obtain the latent posterior of an image and then use the compression algorithm to transmit a sample latent representation from the posterior. Then, the receiver can use the decoder to reconstruct the image from the latent representation they received. Note that this reconstruction contains small errors. In lossless compression, the sender must correct for this and has to transmit the residuals along with the latent code. In lossy compression, we omit the transmission of the residuals, as the model is optimized such that the reconstruction retains high perceptual quality.

---

[*]Equal contribution.

Bits-back methods for lossless compression have been at the center of attention recently. They realize the optimal compression rate postulated by the bits-back argument (Hinton & Van Camp, 1993): For a given image, the optimal compression rate using a latent variable model (such as a VAE) is given by the relative entropy between the latent posterior and the prior $\text{KL}\left[\,q(\mathbf{z}\,|\,\boldsymbol{x})\,||\,p(\mathbf{z})\,\right]$ *plus* the expected residual error $\mathbb{E}\left[-\log P(\boldsymbol{x}\,|\,\mathbf{z})\right]$, where $\boldsymbol{x}$ is the input image, $\mathbf{z}$ denotes the stochastic latent representation, and $p(\mathbf{z})$ is the prior over the latent space. This quantity is also known as the negative Evidence Lower BOund (ELBO).

Current bits-back compression methods use variants of the Bits-Back with Asymmetric Numeral Systems (BB-ANS) algorithm (Townsend et al., 2019, 2020; Ho et al., 2019; Kingma et al., 2019). BB-ANS can achieve the bits-back compression rate asymptotically by allowing the codes in a sequence of images to overlap without losing information (hence getting the bits back). The issue is that the first image requires a string of auxiliary bits to start the sequence, which means that it is inefficient when used to compress a single image. Including the auxiliary bits, the compressed size of a single image is often 2-3 times the original size.[2]

We introduce Relative Entropy Coding (REC), a lossless compression paradigm that subsumes bits-back methods. A REC method can encode a sample from the latent posterior $\boldsymbol{z} \sim q(\mathbf{z}\,|\,\boldsymbol{x})$ with codelength close to the relative entropy $\text{KL}\left[\,q(\mathbf{z}\,|\,\boldsymbol{x})\,||\,p(\mathbf{z})\,\right]$, given a shared source of randomness. Then the residuals are encoded using an entropy coding method such as arithmetic coding (Witten et al., 1987) with codelength $-\log P(\boldsymbol{x}\,|\,\boldsymbol{z})$. This yields a combined codelength close to the negative ELBO in expectation without requiring any auxiliary bits.

We propose a REC method, called index coding (iREC), based on importance sampling. To encode a sample from the posterior $q(\mathbf{z}\,|\,\boldsymbol{x})$, our method relies on a shared sequence of random samples from the prior $\boldsymbol{z}_1, \boldsymbol{z}_2, \cdots \sim p(\mathbf{z})$, which in practice is realised using a pseudo-random number generator with a shared random seed. The algorithm selects an element $\boldsymbol{z}_i$ from the random sequence with high density under the posterior. Then, the code for the sample is simply '$i$', its index in the sequence. Given $i$, the receiver can recover $\boldsymbol{z}_i$ by selecting the $i$th element from the shared random sequence. We show that the codelength of '$i$' is close to the relative entropy $\text{KL}\left[\,q(\mathbf{z}\,|\,\boldsymbol{x})\,||\,p(\mathbf{z})\,\right]$.

Apart from eliminating the requirement for auxiliary bits, REC offers a few further advantages. First, an issue concerning virtually every deep image compression algorithm is that they require a quantized latent space for encoding. This introduces an inherently non-differentiable step to training, which hinders performance and, in some cases, prevents model scaling (Hoogeboom et al., 2019). Since our method relies on a shared sequence of prior samples, it is not necessary to quantize the latent space and it can be applied to off-the-shelf VAE architectures with continuous latent spaces. To our knowledge, our method is the first image compression algorithm that can operate in a continuous latent space.

Second, since the codelength scales with the relative entropy but not the number of latent dimensions, the method is unaffected by the pruned dimensions of the VAE, i.e. dimensions where the posterior collapses back onto the prior (Yang et al., 2020; Lucas et al., 2019) (this advantage is shared with bits-back methods, but not others in general).

Third, our method elegantly extends to lossy compression. If we choose to only encode a sample from the posterior using REC without the residuals, the receiver can use the decoder to reconstruct an image that is close to the original. In this setting, the VAE is optimized using an objective that allows the model to maintain high *perceptual quality* even at low bit rates. The compression rate of the model can be precisely controlled during training using a $\beta$-VAE-like training objective (Higgins et al., 2017). Our empirical results confirm that our REC algorithm is competitive with the state-of-the-art in lossy image compression on the Kodak dataset (Eastman Kodak Company, 1999).

The key contributions of this paper are as follows:

- iREC, a relative entropy coding method that can encode an image with codelength close to the negative ELBO for VAEs. Unlike prior bits-back methods, it does not require auxiliary bits, and hence it is efficient for encoding single images. We empirically confirm these findings on the Cifar10, ImageNet32 and Kodak datasets.
- Our algorithm forgoes the quantization of latent representations entirely, hence it is directly applicable to off-the-shelf VAE architectures with continuous latent spaces.

- The algorithm can be applied to lossy compression, where it is competitive with the state-of-the-art on the Kodak dataset.

## 2 Learned Image Compression

The goal of compression is to communicate some information between the sender and the receiver using as little bandwidth as possible. Lossless compression methods assign some *code* $C(x)$ to some input $x$, consisting of a sequence of bits, such that the original $x$ can be always recovered from $C(x)$. The efficiency of the method is determined by the average length of $C(x)$.

All compression methods operate on the same underlying principle: *Commonly occurring patterns are assigned shorter codelengths while rare patterns have longer codelengths.* This principle was first formalized by Shannon (1948). Given an underlying distribution $P(\mathrm{x})$,[3] where x is the input taking values in some set $\mathcal{X}$, the rate of compression cannot be better than $H[P] = \sum_{x \in \mathcal{X}} -P(x) \log P(x)$, the Shannon-entropy of $P$. This theoretical limit is achieved when the codelength of a given $x$ is close to the negative log-likelihood $|C(x)| \approx -\log P(x)$. Methods that get close to this limit are referred to as entropy coding methods, the most prominent ones being Huffman coding and arithmetic coding (Huffman, 1952; Witten et al., 1987).

The main challenge in image compression is that the distribution $P(\mathrm{x})$ is not readily available. Methods either have to hand-craft $P(\mathrm{x})$ or learn it from data. The appeal in using generative models to learn $P(\mathrm{x})$ is that they can give significantly better approximations than traditional approaches.

### 2.1 Image Compression Using Variational Autoencoders

A Variational Autoencoder (VAE, Kingma & Welling (2014)) is a generative model that learns the underlying distribution of a dataset in an unsupervised manner. It consists of a pair of neural networks called the *encoder* and *decoder*, that are approximate inverses of each other. The encoder network takes an input $\boldsymbol{x}$ and maps it to a posterior distribution over the latent representations $q_\phi(\mathbf{z} \,|\, \boldsymbol{x})$. The decoder network maps a latent representation $\boldsymbol{z} \sim q_\phi(\mathbf{z} \,|\, \boldsymbol{x})$ to the conditional distribution $P_\theta(\mathbf{x} \,|\, \boldsymbol{z})$. Here, $\phi$ and $\theta$ denote the parameters of the encoder and the decoder, respectively. These two networks are trained jointly by maximizing a lower bound to the marginal log-likelihood $\log P(\boldsymbol{x})$, the Evidence Lower BOund $\mathcal{L}(\boldsymbol{x}, \phi, \theta)$ (ELBO):

$$\log P(\boldsymbol{x}) \geq \mathcal{L}(\boldsymbol{x}, \phi, \theta) = \underbrace{\mathbb{E}_{\mathbf{z} \sim q_\phi(\mathbf{z} \,|\, \boldsymbol{x})} \left[ \log P_\theta(\boldsymbol{x} \,|\, \boldsymbol{z}) \right]}_{\text{conditional log-likelihood}} - \underbrace{\mathrm{KL} \left[ q_\phi(\mathbf{z} \,|\, \boldsymbol{x}) \,||\, p(\mathbf{z}) \right]}_{\text{relative entropy}}. \quad (1)$$

The VAE can be used to realize a non-linear form of transform coding (Goyal, 2001) to perform image compression. Given an image $\boldsymbol{x}$, the sender to maps it to its latent posterior $q(\mathbf{z} \,|\, \boldsymbol{x})$, and communicates a sample $\boldsymbol{z} \sim q(\mathbf{z} \,|\, \boldsymbol{x})$ to the receiver (where we omit $\phi$ and $\theta$ for notational ease). Our proposed algorithm can accomplish this with communication cost close to the relative entropy $\mathrm{KL} \left[ q(\mathbf{z} \,|\, \boldsymbol{x}) \,||\, p(\mathbf{z}) \right]$. Given $\boldsymbol{z}$, the receiver can use the decoder to obtain the conditional distribution $P(\mathbf{x} \,|\, \boldsymbol{z})$, which can be used for both lossless and lossy compression. Figure 1 depicts both processes.

For **lossless** compression, the sender can use an entropy coding method with $P(\mathbf{x} \,|\, \boldsymbol{z})$ to encode the residuals. The cost of communicating the residuals is the negative log-likelihood $-\log P(\mathbf{x} \,|\, \boldsymbol{z})$ which yields a combined codelength close to the negative ELBO. (See Figure 33c.)

For **lossy** compression, the most commonly used approach is to take the mean of the conditional distribution to be the approximate reconstruction $\tilde{\boldsymbol{x}} = \mathbb{E}_{\mathbf{x} \sim P(\mathbf{x} \,|\, \boldsymbol{z})} \left[ \mathbf{x} \right]$. This yields a reconstruction close to the original image, while only having to communicate $\boldsymbol{z}$. (See Figure 1b.)

The remaining question is how to communicate a sample $\boldsymbol{z}$ from the posterior $q(\mathbf{z} \,|\, \boldsymbol{x})$ given the shared prior $p(\mathbf{z})$. The most widely given answer is the **quantization** of the latent representations, which are then encoded using entropy coding (Theis et al., 2017; Ballé et al., 2016a). This approach is simple to use but has two key weaknesses. First, because of the quantized latent space, the posterior is a discrete probability distribution, which is significantly more difficult to train with gradient descent than its continuous counterpart. It requires the use of gradient estimators and does not scale well with

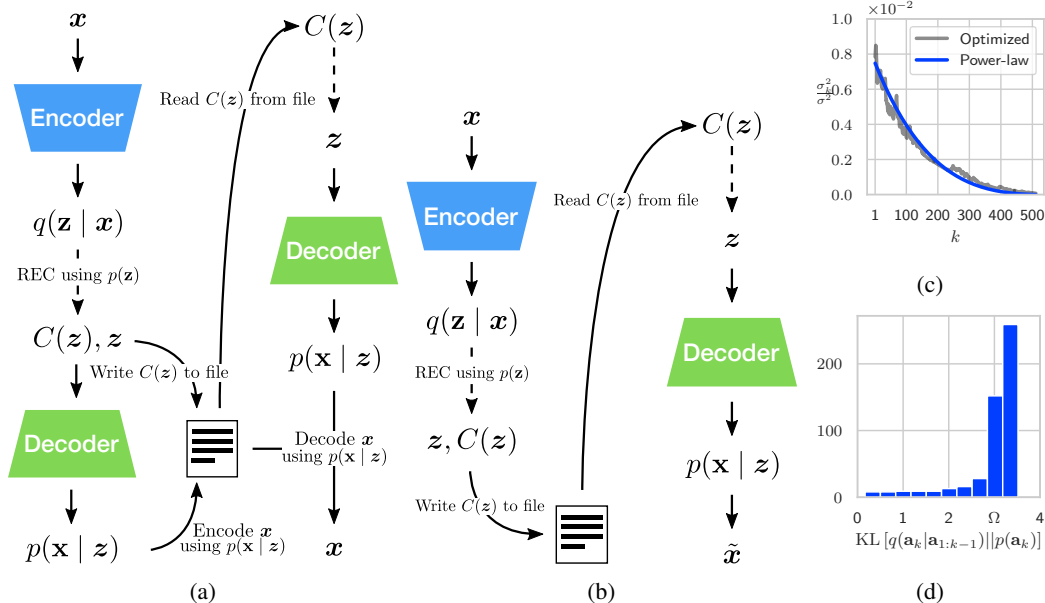

Figure 1: **(a)** Lossless compression using REC **(b)** Lossy compression using REC **(c)** The variances of the coding distributions of the auxiliary variables. We observe that the individually optimized values are well approximated by a power-law. **(d)** The relative entropies of the auxiliary variables are near or below $\Omega$. ((c) and (d) depict statistics from the 23rd stochastic layer of a 24-layer ResNet VAE, since this layer contains the majority of the model's total relative entropy.)

depth (Hoogeboom et al., 2019). A second known shortcoming of this method is that the codelength scales with the number of latent dimensions even when those dimensions as pruned by the VAE, i.e. the posterior coincides with the prior and the dimension, therefore, carries no information (Yang et al., 2020; Lucas et al., 2019).

An alternative to quantization in lossless compression is **bits-back coding** (Townsend et al., 2019). It uses a string of auxiliary bits to encode $z$, which is then followed by encoding the residuals. When compressing multiple images, bits-back coding reuses the code of already compressed images as auxiliary bits to compress the remaining ones, bringing the asymptotic cost close to the negative ELBO. However, due to the use of auxiliary bits, it is inefficient to use for single images.

Relative Entropy Coding (REC), rectifies the aforementioned shortcomings. We propose a REC algorithm that can encode $z$ with codelength close to the relative entropy, without requiring quantization or auxiliary bits. It is effective for both lossy and lossless compression.

## 3 Relative Entropy Coding

Relative Entropy Coding (REC) is a *lossless compression paradigm* that solves the problem of communicating a sample from the posterior distribution $q(\mathbf{z} \mid \mathbf{x})$ given the shared prior distribution $p(\mathbf{z})$. In more general terms, the sender wants to communicate a sample $z \sim q(\mathbf{z})$ to the receiver from a target distribution (the target distribution is only known to the sender) with a coding distribution $p(\mathbf{z})$ shared between the sender and the receiver, given a *shared source of randomness*. That is, over many runs, the empirical distribution of the transmitted $z$s converges to $q(\mathbf{z})$. Hence, REC is a *stochastic* coding scheme, in contrast with entropy coding which is fully deterministic.

We refer to algorithms that achieve communication cost provably close to $\mathrm{KL}\left[\, q(\mathbf{z}) \,\|\, p(\mathbf{z}) \,\right]$ as REC algorithms. We emphasize the counter-intuitive notion, that communicating a stochastic sample from $q(\mathbf{z})$ can be much cheaper than communicating any *specific* sample $z$. For example, consider the case when $p(\mathbf{z}) = q(\mathbf{z}) = \mathcal{N}\left(\mathbf{z} \mid 0, I\right)$. First, consider the naive approach of sampling $z \sim q(\mathbf{z})$ and encoding it with entropy coding. The expected codelength of $z$ is $\infty$ since the Shannon entropy of a Gaussian random variable is $\infty$. Now consider a REC approach: the sender could simply indicate to the receiver that they should draw a sample from the shared coding distribution $p(\mathbf{z})$, which has

$\mathcal{O}(1)$ communication cost. This problem is studied formally in Harsha et al. (2007). They show that the relative entropy is a lower bound to the codelength under mild assumptions. As part of a formal proof, they present a rejection sampling algorithm that in practice is computationally intractable even for small problems, however, it provides a basis for our REC algorithm presented below.

## 3.1 Relative Entropy Coding with Index Coding

We present Index Coding (iREC), a REC algorithm that scales to the needs of modern image compression problems. The core idea for the algorithm is to rely on a shared source of randomness between the sender and the receiver, which takes the form of an infinite sequence of random samples from the prior $z_1, z_2, \cdots \sim p(\mathbf{z})$. This can be practically realized using a pseudo-random number generator with a shared random seed. Then, to communicate an element $z_i$ from the sequence, it is sufficient to transmit its index, i.e. $C(z_i) = i$. From $i$ the receiver can reconstruct $z_i$ by using the shared source of randomness to generate $z_1, z_2, \ldots$ and then selecting the $i$th element.

iREC is based on the importance sampling procedure proposed in Havasi et al. (2019). Let $M = \lceil \exp\left(\mathrm{KL}\left[\,q(\mathbf{z}) \,||\, p(\mathbf{z})\,\right]\right)\rceil$. Our algorithm draws $M$ samples $z_1, \ldots, z_M \sim p(\mathbf{z})$, then selects $z_i$ with probability proportional to the importance weights $P_{\mathrm{accept}}(z_m) \propto \frac{q(z_m)}{p(z_m)}$ for $m \in \{1, \ldots M\}$. Although $z_i$ is not an unbiased sample from $q(\mathbf{z})$, Havasi et al. (2019) show that considering $M$ samples is sufficient to ensure that the bias remains low. The cost of communicating $z_i$ is simply $\log M \approx \mathrm{KL}\left[\,q(\mathbf{z}) \,||\, p(\mathbf{z})\,\right]$, since we only need to communicate $i$ where $1 \le i \le M$.

Importance sampling has promising theoretical properties, however, it is still infeasible to use in practice because $M$ grows exponentially with the relative entropy. To drastically reduce this cost, we propose sampling a sequence of auxiliary variables instead of sampling $q(\mathbf{z})$ directly.

### 3.1.1 Using Auxiliary Variables

We propose breaking $\mathbf{z}$ up into a sequence of $K$ auxiliary random variables $\mathbf{a}_{1:K} = \mathbf{a}_1, \ldots, \mathbf{a}_K$ with independent coding distributions $p(\mathbf{a}_1) \ldots, p(\mathbf{a}_K)$ such that they fully determine $\mathbf{z}$, i.e. $\mathbf{z} = f(\mathbf{a}_{1:K})$ for some function $f$. Our goal is to derive target distributions $q(\mathbf{a}_k \,|\, \mathbf{a}_{1:k-1})$ for each of the auxiliary variables given the previous ones, such that by sampling each of them via importance sampling, i.e. $\mathbf{a}_k \sim q(\mathbf{a}_k \,|\, \mathbf{a}_{1:k-1})$ for $k \in \{1, \ldots K\}$, we get a sample $\mathbf{z} = f(\mathbf{a}_{1:K}) \sim q(\mathbf{z})$. This implies the auxiliary coding distributions $p(\mathbf{a}_k)$ and target distributions $q(\mathbf{a}_k \,|\, \mathbf{a}_{1:k-1})$ must satisfy the marginalization properties

$$p(\mathbf{z}) = \int \delta(f(\boldsymbol{a}_{1:K}) - \mathbf{z})p(\boldsymbol{a}_{1:K})\mathrm{d}\boldsymbol{a}_{1:K} \quad \text{and} \quad q(\mathbf{z}) = \int \delta(f(\boldsymbol{a}_{1:K}) - \mathbf{z})q(\boldsymbol{a}_{1:K})\mathrm{d}\boldsymbol{a}_{1:K}, \quad (2)$$

where $\delta$ is the Dirac delta function, $p(\boldsymbol{a}_{1:K}) = \prod_{k=1}^{K} p(\boldsymbol{a}_k)$ and $q(\boldsymbol{a}_{1:K}) = \prod_{k=1}^{K} q(\boldsymbol{a}_k \,|\, \boldsymbol{a}_{1:k-1})$. Note that the coding distributions $p(\mathbf{a}_1) \ldots, p(\mathbf{a}_K)$ and $f$ can be freely chosen subject to Eq 2.

The cost of encoding each auxiliary variable using importance sampling is equal to the relative entropy between their corresponding target and coding distributions. Hence, to avoid introducing an overhead to the overall codelength, the targets must satisfy $\mathrm{KL}\left[\,q(\mathbf{a}_{1:k}) \,||\, p(\mathbf{a}_{1:k})\,\right] = \mathrm{KL}\left[\,q(\mathbf{z}) \,||\, p(\mathbf{z})\,\right]$.[4] To ensure this, for fixed $f$ and joint auxiliary coding distribution $p(\mathbf{a}_{1:K})$, the joint auxiliary target distributions must have the form

$$q(\mathbf{a}_{1:k}) := \int p(\mathbf{a}_{1:k} \,|\, \boldsymbol{z})q(\boldsymbol{z})\,\mathrm{d}\boldsymbol{z} \quad \text{for } k \in \{1 \ldots K\}. \quad (3)$$

These are the only possible auxiliary targets that satisfy the condition on the sum of the relative entropies, which we formally show in the supplementary material.

**Choosing the forms of the auxiliary variables.** Factorized Gaussian priors are a popular choice for VAEs. For a Gaussian coding distribution $p(\mathbf{z}) = \mathcal{N}(0, \sigma^2 I)$, we propose $\mathbf{z} = \sum_{k=1}^{K} \mathbf{a}_k$, $p(\mathbf{a}_k) = \mathcal{N}(0, \sigma_k^2 I)$ for $k \in \{1, \ldots, K\}$ such that $\sum_{k=1}^{K} \sigma_k^2 = \sigma^2$. The targets $q(\mathbf{a}_k \,|\, \mathbf{a}_{1:k-1})$ in this case turn out to be Gaussian as well, and their form is derived in the supplementary material.

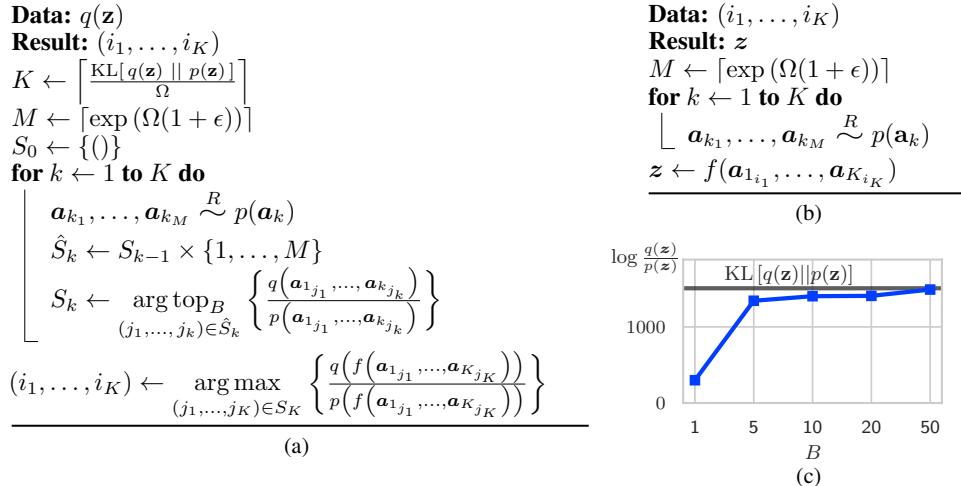

Figure 2: **(a)** iREC encoder **(b)** iREC decoder **(c)** Beam search ensures that $\log \frac{q(\mathbf{z})}{p(\mathbf{z})}$ is close to the relative entropy. $B$ is the number of beams. Plotted using the 23rd stochastic layer of a 24-layer ResNet VAE, since this layer contains the majority of the model's total relative entropy. Here, $\overset{R}{\sim}$ indicates sampling using a pseudo-random number generator with random seed $R$, and $\arg\text{top}_B$ selects the arguments of the top $B$ ranking elements in a set.

To guarantee that every auxiliary variable can be encoded via importance sampling, the relative entropies should be similar across the auxiliary variables, i.e. $\text{KL}\left[q(\mathbf{a}_k \mid \mathbf{a}_{1:k-1}) \mid\mid p(\mathbf{a}_k)\right] \approx \Omega$, where $\Omega$ is a hyperparameter (we used $\Omega = 3$ in our experiments). This yields $K = \lceil \text{KL}\left[q(\mathbf{z}) \mid\mid p(\mathbf{z})\right]/\Omega \rceil$ auxiliary variables in total. We initially set the auxiliary coding distributions by optimizing their variances $\sigma_k^2$ on a small validation set to achieve relative entropies close to $\Omega$. Later, we found that the ratio of the variance $\sigma_k^2$ of the $k$-th auxiliary variable to the remaining variance $\sigma^2 - \sum_{j=1}^{k-1} \sigma_j^2$ is well approximated by the power law $(K + 1 - k)^{-0.79}$, as shown in Figure 1c. In practice, we used this approximation to set each $\sigma_k^2$. With these auxiliary variables, we fulfil the requirement of keeping the individual relative entropies near or below $\Omega$ as shown empirically in Figure 1d. To account for the auxiliary variables whose relative entropy slightly exceeds $\Omega$, in practice, we draw $M = \lceil \exp\left(\Omega(1 + \epsilon)\right) \rceil$ samples, where $\epsilon$ is a small non-negative constant (we used $\epsilon = 0.2$ for lossless and $\epsilon = 0.0$ for lossy compression), leading to a $(1 + \epsilon)$ increase of the codelength.

### 3.1.2 Reducing the Bias with Beam Search

An issue with naively applying the auxiliary variable scheme is that the cumulative bias of importance sampling each auxiliary variable adversely impacts the compression performance, potentially leading to higher distortion and hence longer codelength.

To reduce this bias, we propose using a beam search algorithm (shown in Figure 2) to search over multiple possible assignments of the auxiliary variables. We maintain a set of the $B$ lowest bias samples from the auxiliary variables and used the log-importance weight $\log \frac{q(\mathbf{z})}{p(\mathbf{z})}$ as a heuristic measurement of the bias. For an unbiased sample $\log \frac{q(\mathbf{z})}{p(\mathbf{z})} \approx \text{KL}\left[q(\mathbf{z}) \mid\mid p(\mathbf{z})\right]$, but for a biased sample, $\log \frac{q(\mathbf{z})}{p(\mathbf{z})} \ll \text{KL}\left[q(\mathbf{z}) \mid\mid p(\mathbf{z})\right]$. For each auxiliary variable $\mathbf{a}_k$ in the sequence, we combine the $B$ lowest bias samples for $\mathbf{a}_{1:k-1}$ with the $M$ possible importance samples for $\mathbf{a}_k$. To choose the $B$ lowest bias samples for the next iteration, we take the top $B$ samples with the highest importance weights $\frac{q(\mathbf{a}_{1:k})}{p(\mathbf{a}_{1:k})}$ out of the $B \times M$ possibilities. At the end, we select $\mathbf{a}_{1:K}$ with the highest importance weight $\frac{q(f(\mathbf{a}_{1:K}))}{p(f(\mathbf{a}_{1:K}))} = \frac{q(\mathbf{z})}{p(\mathbf{z})}$.

### 3.1.3 Determining the hyperparameters

Finding good values for $\Omega$, $\epsilon$ and $B$ is crucial for the good performance of our method. We want as short a codelength as possible, while also minimizing computational cost. Therefore, we ran a grid search over a reasonable range of parameter settings for the lossless compression of a small number of ImageNet32 images. We compare the efficiency of settings by measuring the codelength overhead they produced in comparison to the ELBO, which represents optimal performance. We find that for reasonable settings of $\Omega$ (between 5-3) and for fixed $\epsilon$, regular importance sampling ($B = 1$) gives between 25-80% overhead, whereas beam search with $B = 5$ gives 15-25%, and with $B = 20$ it gives 10-15%. Setting $B > 20$ does not result in significant improvements in overhead, while the computational cost is heavily increased. Thus, we find that 10-20 beams are sufficient to significantly reduce the bias as shown in Figure 2c. The details of our experimental setup and the complete report of our findings can be found in the supplementary material.

## 4 Experiments

We compare our method against state-of-the-art lossless and lossy compression methods. Our experiments are implemented in `TensorFlow` (Abadi et al., 2015) and are publicly available at `https://github.com/gergely-flamich/relative-entropy-coding`.

### 4.1 Lossless Compression

We compare our method on single image lossless compression (shown in Table 1) against PNG, WebP and FLIF, Integer Discrete-Flows (Hoogeboom et al., 2019) and the prominent bits-back approaches: Local Bits-Back Coding (Ho et al., 2019), BitSwap (Kingma et al., 2019) and HiLLoC (Townsend et al., 2020).

Our model for these experiments is a ResNet VAE (RVAE) (Kingma et al., 2016) with 24 Gaussian stochastic levels. This model utilizes skip-connections to prevent the posteriors on the higher stochastic levels to collapse onto the prior and achieves an ELBO that is competitive with the current state-of-the-art auto-regressive models. For better comparison, we used the exact model used by Townsend et al. (2020)[5] trained on ImageNet32. The three hyperparameters of iREC are set to $\Omega = 3$, $\epsilon = 0.2$ and $B = 20$. Further details on our hyperparameter tuning process are included in the supplementary material.

We evaluated the methods on Cifar10 and ImageNet32 comprised of $32 \times 32$ images, and the Kodak dataset comprised of full-sized images. For Cifar10 and ImageNet32, we used a subsampled test set of size 1000 due to the speed limitation of our method (currently, it takes $\sim 1$ minute to compress a $32 \times 32$ image and 1-10 minutes to compress a large image). By contrast, decoding with our method is fast since it does not require running the beam search procedure.

iREC significantly outperforms other bits-back methods on all datasets since it does not require auxiliary bits, although it is still slightly behind non-bits-back methods as it has a $\sim 20\%$ overhead compared to the ELBO due to using $\epsilon = 0.2$.

### 4.2 Lossy Compression

On the lossy compression task, we present average rate-distortion curves calculated using the PSNR (Huynh-Thu & Ghanbari, 2008) and MS-SSIM (Wang et al., 2004) quality metrics on the Kodak dataset, shown in Figure 3. On both metrics, we compare against JPEG, BPG, Theis et al. (2017) and Ballé et al. (2018), with the first two being classical methods and the latter two being ML-based. Additionally, on PSNR we compare against Minnen & Singh (2020), whose work represents the current state-of-the-art to the best of our knowledge.[7]

Table 1: Single image, lossless compression performance in bits per dimension (lower is better). The best performing bits-back or REC method is highlighted for each dataset. The asymptotic rates are included in parenthesis where they are different from the single image case. To calculate the number of bits needed for single images, we added the number of auxiliary bits required to the asymptotic compression rate as reported in the respective papers.

|  |  | Cifar10 (32x32) | ImageNet32 (32x32) | Kodak (768x512) |
|---|---|---|---|---|
| *Non bits-back* | PNG | 5.87 | 6.39 | 4.35 |
|  | WebP | 4.61 | 5.29 | 3.20 |
|  | FLIF | 4.19 | 4.52 | 2.90 |
|  | IDF | 3.34 | 4.18 | – |
| *Bits-back*[6] | LBB | 54.96 (3.12) | 55.72 (3.88) | – |
|  | BitSwap | 6.53 (3.82) | 6.97 (4.50) | – |
|  | HiLLoC | 24.51 (3.56) | 26.80 (4.20) | 17.5 (3.00) |
| *REC* | iREC (Ours) | **4.18** | **4.91** | **3.67** |
|  | ELBO (RVAE) | [3.55] | [4.18] | [3.00] |

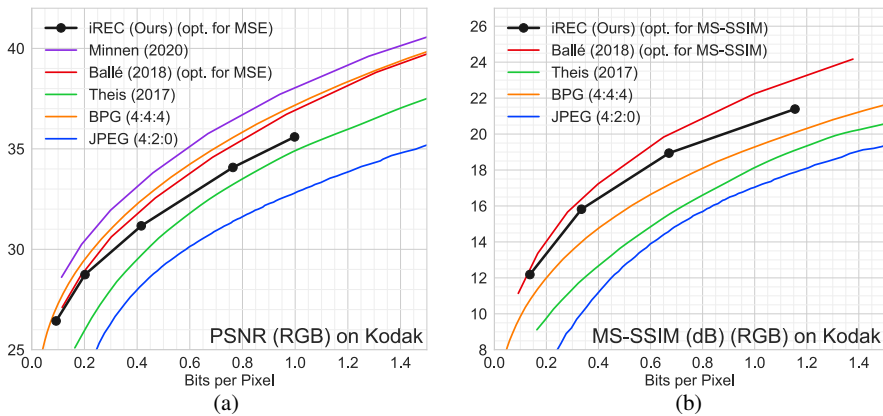

Figure 3: Comparison of REC against classical methods such as JPEG, BPG and competing ML-based methods. **(a)** PSNR comparisons **(b)** MS-SSIM comparisons in decibels, calculated using the formula $-10 \log_{10}(1 - \text{MS-SSIM})$. See the supplementary material for more comparisons.

We used the architecture presented in Ballé et al. (2018) with the latent distributions changed to Gaussians and a few small modifications to accommodate this; see the supplementary material for precise details. Following Ballé et al. (2018), we trained several models using $\mathcal{L}_\lambda(\boldsymbol{x}, \phi, \theta) = \lambda D(\boldsymbol{x}, \hat{\boldsymbol{x}}) - \text{KL}\left[q_\phi(\mathbf{z} \mid \boldsymbol{x}) \mid\mid p(\mathbf{z})\right]$, where $\hat{\boldsymbol{x}}$ is the reconstruction of the image $\boldsymbol{x}$, and $D(\cdot, \cdot) \in \{\text{MSE}, \text{MS-SSIM}\}$ is a differentiable distortion metric.[8] Varying $\lambda$ in the loss yields models with different rate-distortion trade-offs. We optimized 5 models for MSE with $\lambda \in \{0.001, 0.003, 0.01, 0.03, 0.05\}$ and 4 models for MS-SSIM with $\lambda \in \{0.003, 0.01, 0.03, 0.08\}$. The hyperparameters of iREC were set this time to $\Omega = 3$, $\epsilon = 0$ and $B = 10$. As can be seen in Figure 3, iREC is competitive with the state-of-the-art lossy compression methods on both metrics.

## 5 Related Work

Multiple recent bits-back papers build on the work of Townsend et al. (2019), who first proposed BB-ANS. These works elevate the idea from a theoretical argument to a practically applicable algorithm. Kingma et al. (2019) propose BitSwap, an improvement to BB-ANS that allows it to be applied to

hierarchical VAEs. Ho et al. (2019) extend the work to general flow models, which significantly improves the asymptotic compression rate. Finally, Townsend et al. (2020) apply the original BB-ANS algorithm to full-sized images and improve its run-time by vectorizing its implementation.

All previous VAE-based lossy compression methods use entropy coding with quantized latent representations. They propose different approaches to circumvent the non-differentiability that this presents during training. Most prominently, Ballé et al. (2016a) describe a continuous relaxation of quantization based on dithering. Building on this idea, Ballé et al. (2018) introduce a 2-level hierarchical architecture, and Minnen et al. (2018); Lee et al. (2019); Minnen & Singh (2020) explore more expressive latent representations, such as using learned adaptive context models to aid their entropy coder, and autoregressive latent distributions.

## 6 Conclusion

This paper presents iREC, a REC algorithm, which extends the importance sampler proposed by Havasi et al. (2019). It enables the use of latent variable models (and VAEs in particular) with continuous probability distributions for both lossless and lossy compression.

Our method significantly outperforms bits-back methods on lossless single image compression benchmarks and is competitive with the asymptotic performance of competing methods. On the lossy compression benchmarks, our method is competitive with the state-of-the-art for both the PSNR and MS-SSIM perceptual quality metrics. Currently, the main practical limitation of bits-back methods, including our method, is the compression speed, which we hope to improve in future works.

## 7 Acknowledgements

Marton Havasi is funded by EPSRC. We would like to thank Stratis Markou and Ellen Jiang for the helpful comments on the manuscript.

## 8 Broader Impact

Our work presents a novel data compression framework and hence inherits both its up and downsides. In terms of positive societal impacts, data compression reduces the bandwidth requirements for many applications and websites, making them more inexpensive to access. This increases accessibility to online content in rural areas with limited connectivity or underdeveloped infrastructure. Moreover, it reduces the energy requirement and hence the environmental impact of information processing systems. However, care must be taken when storing information in a compressed form for long time periods, and backwards-compatibility of decoders must be maintained, as data may otherwise be irrevocably lost, leading to what has been termed the Digital Dark Ages (Kuny, 1997).

## Footnotes

[2]Based on the number of auxiliary bits recommended by Townsend et al. (2020).

[3]In this paper use roman letters (e.g. $\mathbf{x}$) for random variables and italicized letters (e.g. $\boldsymbol{x}$) for their realizations. Bold letters denote vectors. By a slight abuse of notation, we denote $p(\mathbf{x} = \boldsymbol{x})$ by $p(\boldsymbol{x})$.

[4] We use here the definition $\mathrm{KL}\left[\,q(\mathrm{x} \,|\, \mathrm{y}) \,||\, p(\mathrm{x})\,\right] = \mathbb{E}_{x,y \sim p(\mathrm{x,y})}\left[\log \frac{q(\mathrm{x} \,|\, \mathrm{y})}{p(\mathrm{x})}\right]$ (Cover & Thomas, 2012).

[5]We used the publicly available trained weights published by the authors of Townsend et al. (2020).

[6]To overcome the issue of the inefficiency of bits-back methods for single or small-batch image compression, in practice an efficient single-image compression method (e.g. FLIF) is used to encode the first few images and only once the overhead of using bits-back methods becomes negligible do we switch to using them (see e.g. Townsend et al. (2020)).

[7]For all competing methods, we used publicly available data at `https://github.com/tensorflow/compression/tree/master/results/image_compression`.

[8]In practice we used $1 - \text{MS-SSIM}$ as the loss function with power factors set to $\alpha = \beta = \gamma = 1$.

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
