[Supplementary Material]

# A    Deriving the posterior distributions for the auxiliary variables

In this section, we derive Equation (3) presented in the main text, and show how this general result can be applied to the case where $\mathbf{z}$ and all $\mathbf{a}_k$s are selected to be Gaussian, and $f(\mathbf{a}_{1:K}) = \sum_{k=1}^{K} \mathbf{a}_k$, and show a simple scheme to implement the proposed auxiliary variable method in practice.

## A.1    Deriving the $q(\mathbf{a}_{1:k})$s - general case

Here we show that the form of the auxiliary posterior $q(\mathbf{a}_{1:k})$ presented in Equation (3) is the only suitable choice such that the KL divergence remains unchanged. Concretely, fix $f$ and the auxiliary coding distributions $p(\mathbf{a}_k \,|\, \mathbf{a}_{1:k-1})$ for $k \in \{1, \ldots, K\}$. Now, we want to find $q(\mathbf{a}_{1:K})$ such that the condition

$$\mathrm{KL}\left[\, q(\mathbf{a}_{1:K}) \,||\, p(\mathbf{a}_{1:K}) \,\right] = \mathrm{KL}\left[\, q(\mathbf{z}) \,||\, p(\mathbf{z}) \,\right] \tag{4}$$

is satisfied. Observe that

$$\begin{aligned}
\mathrm{KL}\left[\, q(\mathbf{z}, \mathbf{a}_{1:K}) \,||\, p(\mathbf{z}, \mathbf{a}_{1:K}) \,\right] &= \mathrm{KL}\left[\, q(\mathbf{z}) \,||\, p(\mathbf{z}) \,\right] + \mathrm{KL}\left[\, q(\mathbf{a}_{1:K} \,|\, \mathbf{z}) \,||\, p(\mathbf{a}_{1:K} \,|\, \mathbf{z}) \,\right] \\
&= \mathrm{KL}\left[\, q(\mathbf{a}_{1:K}) \,||\, p(\mathbf{a}_{1:K}) \,\right] + \mathrm{KL}\left[\, q(\mathbf{z} \,|\, \mathbf{a}_{1:K}) \,||\, p(\mathbf{z} \,|\, \mathbf{a}_{1:K}) \,\right],
\end{aligned} \tag{5}$$

where the two equalities follow from breaking up the joint KL using the chain rule of relative entropies in two different ways. Notice, that since $\mathbf{z} = f(\mathbf{a}_{1:K})$ is a deterministic relationship, $q(\mathbf{z} \,|\, \mathbf{a}_{1:K}) = p(\mathbf{z} \,|\, \mathbf{a}_{1:K}) = \delta\left(f(\mathbf{a}_{1:K}) - \mathbf{z}\right)$. Hence, we have $\mathrm{KL}\left[\, q(\mathbf{z} \,|\, \mathbf{a}_{1:K}) \,||\, p(\mathbf{z} \,|\, \mathbf{a}_{1:K}) \,\right] = 0$. Using this fact to simplify Eq 5, we get

$$\mathrm{KL}\left[\, q(\mathbf{z}) \,||\, p(\mathbf{z}) \,\right] + \mathrm{KL}\left[\, q(\mathbf{a}_{1:K} \,|\, \mathbf{z}) \,||\, p(\mathbf{a}_{1:K} \,|\, \mathbf{z}) \,\right] = \mathrm{KL}\left[\, q(\mathbf{a}_{1:K}) \,||\, p(\mathbf{a}_{1:K}) \,\right]. \tag{6}$$

We see that our original condition in Eq 4 for the auxiliary targets is satisfied when $\mathrm{KL}\left[\, q(\mathbf{a}_{1:K} \,|\, \mathbf{z}) \,||\, p(\mathbf{a}_{1:K} \,|\, \mathbf{z}) \,\right] = 0$. Applying the chain rule of relative entropies $K$ times, the condition can be rewritten as

$$\mathrm{KL}\left[\, q(\mathbf{a}_{1:K} \,|\, \mathbf{z}) \,||\, p(\mathbf{a}_{1:K} \,|\, \mathbf{z}) \,\right] = \sum_{k=1}^{K} \mathrm{KL}\left[\, q(\mathbf{a}_k \,|\, \mathbf{a}_{1:k-1}, \mathbf{z}) \,||\, p(\mathbf{a}_k \,|\, \mathbf{a}_{1:k-1}, \mathbf{z}) \,\right] = 0. \tag{7}$$

Due to the non-negativity of relative entropy, the above is satisfied if and only if $\mathrm{KL}\left[\, q(\mathbf{a}_k \,|\, \mathbf{a}_{1:k-1}, \mathbf{z}) \,||\, p(\mathbf{a}_k \,|\, \mathbf{a}_{1:k-1}, \mathbf{z}) \,\right] = 0 \,\forall\, k \in \{1, \ldots, K\}$. This further implies, that

$$q(\mathbf{a}_k \,|\, \mathbf{a}_{1:k-1}, \mathbf{z}) = p(\mathbf{a}_k \,|\, \mathbf{a}_{1:k-1}, \mathbf{z}) \quad \text{as well as} \quad q(\mathbf{a}_{1:k} \,|\, \mathbf{z}) = p(\mathbf{a}_{1:k} \,|\, \mathbf{z}) \tag{8}$$

for all $k \in \{1, \ldots, K\}$. Now, fix $k$. Then, we have

$$\begin{aligned}
q(\mathbf{a}_{1:k}) &= \int q(\mathbf{a}_{1:k} \,|\, \mathbf{z}) q(\mathbf{z}) \mathrm{d}\mathbf{z} \\
&= \int p(\mathbf{a}_{1:k} \,|\, \mathbf{z}) q(\mathbf{z}) \mathrm{d}\mathbf{z}
\end{aligned} \tag{9}$$

by Eq 8, as required.

## A.2    An iterative scheme

A simple way to directly implement an auxiliary variable coding scheme described in Section 3.1.1 is to draw samples sequentially from the conditional auxiliary targets $q(\mathbf{a}_k \,|\, \mathbf{a}_{1:k-1})$. Here, we note the following pair recursive relationships:

$$\begin{aligned}
q(\mathbf{a}_k \,|\, \mathbf{a}_{1:k-1}) &= \int q(\mathbf{a}_k \,|\, \mathbf{a}_{1:k-1}, \mathbf{z}) q(\mathbf{z} \,|\, \mathbf{a}_{1:k-1}) \mathrm{d}\mathbf{z} \\
&= \int p(\mathbf{a}_k \,|\, \mathbf{a}_{1:k-1}, \mathbf{z}) q(\mathbf{z} \,|\, \mathbf{a}_{1:k-1}) \mathrm{d}\mathbf{z}
\end{aligned} \tag{10}$$

$$q(\mathbf{z} \,|\, \mathbf{a}_{1:k}) = \frac{q(\mathbf{z}, \mathbf{a}_k \,|\, \mathbf{a}_{1:k-1})}{q(\mathbf{a}_k \,|\, \mathbf{a}_{1:k-1})} = \frac{p(\mathbf{a}_k \,|\, \mathbf{a}_{1:k-1}, \mathbf{z}) q(\mathbf{z} \,|\, \mathbf{a}_{1:k-1})}{q(\mathbf{a}_k \,|\, \mathbf{a}_{1:k-1})},$$

where the second equality in both identities follows from Eq 8. We can therefore proceed to sequentially encode the $\mathbf{a}_k$s using Algorithm 1.

**Algorithm 1:** Simple sequential auxiliary coding scheme. The iREC$(\cdot, \cdot)$ function performs index coding as described in Section 3.1.

---

**Data:** $q(\mathbf{z}), p(\mathbf{a}_k \,|\, \mathbf{a}_{1:k-1}) \quad \forall k \in \{1, \dots, K\}$
**Result:** $S = (i_1, \dots, i_K)$
$S \leftarrow ()$
$q_0(\mathbf{z}) \leftarrow q(\mathbf{z})$
**for** $k \leftarrow 1$ **to** $K$ **do**

$\quad q(\mathbf{a}_k \,|\, \mathbf{a}_{1:k-1}) \leftarrow \int p(\mathbf{a}_k \,|\, \mathbf{a}_{1:k-1}, \boldsymbol{z}) q_{k-1}(\boldsymbol{z}) \mathrm{d}\boldsymbol{z}$

$\quad i_k, \boldsymbol{a}_k \leftarrow \mathrm{iREC}(q(\mathbf{a}_k \,|\, \mathbf{a}_{1:k-1}), p(\mathbf{a}_k \,|\, \mathbf{a}_{1:k-1}))$
$\quad$ Append $i_k$ to $S$

$\quad q_k(\mathbf{z}) \leftarrow \frac{p(\mathbf{a}_k \,|\, \mathbf{a}_{1:k-1}, \boldsymbol{z}) q_{k-1}(\mathbf{z})}{q(\mathbf{a}_k \,|\, \mathbf{a}_{1:k-1})}$

---

### A.3 The independent Gaussian sum case

In this section we derive the form of the auxiliary coding targets $q(\mathbf{a}_k \,|\, \mathbf{a}_{1:k-1})$ and the conditionals $q(\mathbf{z} \,|\, \mathbf{a}_k)$ in the case when $\mathbf{a}_k$ are all independent and Gaussian distributed, and $\mathbf{z} = f(\mathbf{a}_{1:K}) = \sum_{k=1}^{K} \mathbf{a}_k$. We only derive the result in the univariate case, extending to the diagonal covariance case is straight forward. However, to keep notation consistent with the rest of the paper, only in this section we will keep using the boldface symbols to denote the random quantities and their realizations.

First, let $p(\mathbf{a}_k) = \mathcal{N}\left(\mathbf{a}_k \,|\, \mu_k, \sigma_k^2\right)$. Now, define $\mathbf{b}_k = \sum_{i=1}^{k} \mathbf{a}_i$, $m_k = \sum_{i=K-k+1}^{K} \mu_i$ and $s_k^2 = \sum_{i=K-k+1}^{K} \sigma_k^2$. Then, since $\mathbf{z} = \sum_{k=1}^{K} \mathbf{a}_k$, using the formula for the conditional distribution of sums of Gaussian random variables[9], we get

$$p(\mathbf{a}_k \,|\, \mathbf{a}_{1:k-1}, \mathbf{z}) = \mathcal{N}\left(\mathbf{a}_k \,\middle|\, \mu_k + (\mathbf{z} - \mathbf{b}_{k-1} - m_{k-1})\frac{\sigma_k^2}{s_k^2}, \frac{s_{k+1}^2 \sigma_k^2}{s_k^2}\right). \tag{11}$$

Assume that we have already calculated $q(\mathbf{z} \,|\, \mathbf{a}_{1:k-1}) = \mathcal{N}\left(\mathbf{z} \,|\, \nu_{k-1}, \rho_{k-1}^2\right)$. From here, we notice that by Eq 10, both quantities of interest are products and integrals of Gaussian densities, and hence after some algebraic manipulation, we get

$$q(\mathbf{a}_k \,|\, \mathbf{a}_{1:k-1}) = \mathcal{N}\left(\mathbf{a}_k \,\middle|\, \mu_k + (\nu_{k-1} - \mathbf{b}_{k-1} - m_{k-1})\frac{\sigma_k^2}{s_k^2}, \frac{s_{k+1}^2 \sigma_k^2}{s_k^2} + \rho_{k-1}^2 \frac{\sigma_k^4}{s_k^4}\right), \tag{12}$$

and

$$q(\mathbf{z} \,|\, \mathbf{a}_{1:k}) = \mathcal{N}\left(\mathbf{z} \,\middle|\, \frac{(\mathbf{a}_k - \mu_k)\rho_{k-1}^2 s_k^2 + (\mathbf{b}_{k-1} + m_{k-1})\sigma_k^2 \rho_{k-1}^2 + \nu_{k-1} s_{k+1}^2 s_k^2}{\sigma_k^2 \rho_{k-1}^2 + s_k^2 s_{k+1}^2}, \frac{\rho_{k-1}^2 s_k^2 s_{k+1}^2}{\sigma_k^2 \rho_{k-1}^2 + s_k^2 s_{k+1}^2}\right). \tag{13}$$

Given the above two formulae, both the simple iterative scheme presented in the previous section as well as the beam search algorithm presented in the main text can be readily implemented.

## B Hyperparameter Experiments for Beam Search

Our lossless compression approach has three hyperparameters: $\Omega$, $\epsilon$ and $B$. We tuned these on a small validation set of 10 images from ImageNet32 by sweeping them, and measuring the performance. The codelength can get arbitrarly close to the ELBO, but it requires a significant computational cost. We try to find a good balance between the codelength and the computational cost.

Figure 4 shows the parameter combinations that we tested. We plot 4 metrics:

- 1st row: Overhead in number of bits required compared to the ELBO. A value of 0.2 corresponds to 20% overhead over the ELBO in codelength.

- 2nd row: Time it takes to run the method in seconds.

- 3rd row: Residual overhead. The overhead in codelength when only looking at the residual. This helps to estimate the bias in the samples, since if there is no bias, this overhead should be 0.

- 4th row: For how many out of the 10 validation images did the method crash. Every crash was cause by memory overflow.

Figure 5 depicts the same data, but plotted in two dimensions: overhead vs time.

## C   Auxiliary variables

We plotted the 23rd layer of the RVAE in the main paper to demonstrate how the auxiliary variables are able to break up the latent variables such that their relative entropies are close to $\Omega$. Here we include all 24 layers (Figure 6).

## D   Additional Information on the Setup of Lossy Experiments

We used an appropriately modified version of the architecture proposed by Ballé et al. (2018). Concretely, we swapped all latent distributions for Gaussians, and made the encoder and decoder networks two-headed on the appropriate layers to provide both a mean and log-standard deviation prediction for the latent distributions. For a depiction of our architecture, see Figure 7.

During training, we inferred the parameters of the hyperprior as well (Empirical Bayes). We trained every model on the CLIC 2018 dataset for $2 \times 10^5$ iterations with a batch size of 8 image patches. As done in Ballé et al. (2018), the patches were $256 \times 256$ and were randomly cropped from the training images. As the dataset is curated for lossy image compression tasks, we performed no further data preprocessing or augmentation. We found that annealing the KL divergence in the beginning (also known as warm-up) did not yield a significant performance increase.

## E   Additional Lossy Compression Results

In this section we present some additional results that clarify the current shortcomings of iREC and better illustrate its performance on individual images.

First, we present an extended version of Figure 3 from the main text in Figure 8.

Figure 4: Hyperparameters for lossless compression. $\Omega$ is referred as 'KL_per_partition', $\epsilon$ is referred as 'extra_samples' and $B$ is referred as 'n_beams'. (On a computer, zoom in to see precise figures)

Figure 5: Pareto frontier of the hyperparameters.

## E.1 Actual vs Ideal Performance Comparison

Since iREC is based on importance sampling, the posterior sample it returns will be slightly biased, which affects the distortion of the reconstruction. Furthermore, since it might require setting the oversampling rate $\epsilon > 0$ in some cases, as well as having to communicate some minimal additional side information, the codelength will also be slightly higher than the theoretical lower bound.

We quantify these discrepancies through visualizing *actual* and *ideal* aggregate rate-distortion curves on the Kodak dataset in Figure 9. Concretely, we calculate the actual performance as in the main text, i.e. the bits per pixel are simply calculated from the compressed file size, and the distortion is calculated using the slightly biased sample given by iREC. The ideal bits per pixel is calculated by dividing the $\mathrm{KL}\left[\,q(\mathbf{z}\,|\,\boldsymbol{x})\,\|\,p(\mathbf{z})\,\right]$ by the number of pixels, and the ideal distortion is calculated using a sample drawn from $q(\mathbf{z}\,|\,\boldsymbol{x})$.

As we can see, the distortion gap increases in low-distortion regimes. This is unsurprising, since a low-distortion model's decoder will be more sensitive to biased samples. Interestingly, the ideal performance of our model matches the performance of the method of Ballé et al. (2018), even though they used a much more flexible non-parametric hyperprior and their priors and posteriors were picked to suit image compression. On the other hand, our model only used diagonal Gaussian distributions everywhere.

(a) Performance on PSNR.

(b) Performance on MS-SSIM.

Figure 9: Actual vs Ideal Performance Comparison

## E.2 Performance Comparisons on Individual Kodak Images

Aggregate rate-distortion curves can only serve as a way to compare competing methods, and cannot be used to assess absolute method performance. To address this, we present performance comparisons on individual Kodak images juxtaposed with the images and their reconstructions.

Figure 6: Histograms of the relative entropies of the auxiliary variables in a 24 layer RVAE.

Output

Input

First Level

Sigmoid

Conv2D 5x5x3 / u2

IGDN

Conv2D 5x5xN / u2

IGDN

Conv2D 5x5xN / u2

IGDN

Conv2D 5x5xN / u2

Conv2D 5x5xN / d2

GDN

Conv2D 5x5xN / d2

GDN

Conv2D 5x5xN / d2

GDN

Conv2D 5x5xF / d2

Conv2D 5x5xF / d2

Exp

Second Level

Conv2D 5x5xF

Exp

Conv2D 5x5xF

Leaky ReLU

Conv2D 5x5xM

Leaky ReLU

Conv2D 5x5xM / u2

Conv2D 5x5xM

Leaky ReLU

Conv2D 5x5xM

Leaky ReLU

Conv2D 5x5xG / d2

Conv2D 5x5xG / d2

Sigmoid

Blocks / Layers

Input / Output

Shared for posterior and prior prediction

Posterior prediction

Image reconstruction

Arrows

Output of source fed to input of target

Sample from source fed to input of target

Output of source predicts mean of target

Output of source predicts variance of target

Distributions

Priors

Posteriors

Figure 7: PLN network architecture. The blocks signal data transformations, the arrows signal the flow of information. **Block descriptions:** *Conv2D:* 2D convolutions along the spatial dimensions, where the $W \times H \times C/S$ implies a $W \times H$ convolution kernel, with $C$ target channels and $S$ gives the downsampling rate (given a preceding letter "d") or the upsampling rate (given a preceding letter "u"). If the slash is missing, it means that there is no up/downsampling. All convolutions operate in `same` mode with zero-padding. *GDN / IGDN:* these are the non-linearities described in Ballé et al. (2016a). *Leaky ReLU:* elementwise non-linearity defined as $\max\{x, \alpha x\}$, where we set $\alpha = 0.2$. *Sigmoid:* Elementwise non-linearity defined as $\frac{1}{1+\exp\{-x\}}$.

(a) MS-SSIM rate-distortion curve for Kodak image 1.

(b) PSNR rate-distortion curve for Kodak image 1.

(c) Kodak image 1: Original

(d) Kodak image 1: Reconstruction using model trained with $\lambda = 0.01$

(a) MS-SSIM rate-distortion curve for Kodak image 2.

(b) PSNR rate-distortion curve for Kodak image 2.

(c) Kodak image 2: Original

(d) Kodak image 2: Reconstruction using model trained with $\lambda = 0.01$

(a) MS-SSIM rate-distortion curve for Kodak image 3.

(b) PSNR rate-distortion curve for Kodak image 3.

(c) Kodak image 3: Original

(d) Kodak image 3: Reconstruction using model trained with $\lambda = 0.01$

(a) MS-SSIM rate-distortion curve for Kodak image 4.

(b) PSNR rate-distortion curve for Kodak image 4.

(c) Kodak image 4: Original

(d) Kodak image 4: Reconstruction using model trained with $\lambda = 0.01$

(a) MS-SSIM rate-distortion curve for Kodak image 5.

(b) PSNR rate-distortion curve for Kodak image 5.

(c) Kodak image 5: Original

(d) Kodak image 5: Reconstruction using model trained with $\lambda = 0.01$

(a) MS-SSIM rate-distortion curve for Kodak image 6.

(b) PSNR rate-distortion curve for Kodak image 6.

(c) Kodak image 6: Original

(d) Kodak image 6: Reconstruction using model trained with $\lambda = 0.01$

(a) MS-SSIM rate-distortion curve for Kodak image 7.

(b) PSNR rate-distortion curve for Kodak image 7.

(c) Kodak image 7: Original

(d) Kodak image 7: Reconstruction using model trained with $\lambda = 0.01$

(a) MS-SSIM rate-distortion curve for Kodak image 8.

(b) PSNR rate-distortion curve for Kodak image 8.

(c) Kodak image 8: Original

(d) Kodak image 8: Reconstruction using model trained with $\lambda = 0.01$

(a) MS-SSIM rate-distortion curve for Kodak image 9.

(b) PSNR rate-distortion curve for Kodak image 9.

(c) Kodak image 9: Original

(d) Kodak image 9: Reconstruction using model trained with $\lambda = 0.01$

(a) MS-SSIM rate-distortion curve for Kodak image 10.

(b) PSNR rate-distortion curve for Kodak image 10.

(c) Kodak image 10: Original

(d) Kodak image 10: Reconstruction using model trained with $\lambda = 0.01$

(a) MS-SSIM rate-distortion curve for Kodak image 11.

(b) PSNR rate-distortion curve for Kodak image 11.

(c) Kodak image 11: Original

(d) Kodak image 11: Reconstruction using model trained with $\lambda = 0.01$

(a) MS-SSIM rate-distortion curve for Kodak image 12.

(b) PSNR rate-distortion curve for Kodak image 12.

(c) Kodak image 12: Original

(d) Kodak image 12: Reconstruction using model trained with $\lambda = 0.01$

(a) MS-SSIM rate-distortion curve for Kodak image 13.

(b) PSNR rate-distortion curve for Kodak image 13.

(c) Kodak image 13: Original

(d) Kodak image 13: Reconstruction using model trained with $\lambda = 0.01$

(a) MS-SSIM rate-distortion curve for Kodak image 14.

(b) PSNR rate-distortion curve for Kodak image 14.

(c) Kodak image 14: Original

(d) Kodak image 14: Reconstruction using model trained with $\lambda = 0.01$

(a) MS-SSIM rate-distortion curve for Kodak image 15.

(b) PSNR rate-distortion curve for Kodak image 15.

(c) Kodak image 15: Original

(d) Kodak image 15: Reconstruction using model trained with $\lambda = 0.01$

(a) MS-SSIM rate-distortion curve for Kodak image 16.

(b) PSNR rate-distortion curve for Kodak image 16.

(c) Kodak image 16: Original

(d) Kodak image 16: Reconstruction using model trained with $\lambda = 0.01$

(a) MS-SSIM rate-distortion curve for Kodak image 17.

(b) PSNR rate-distortion curve for Kodak image 17.

(c) Kodak image 17: Original

(d) Kodak image 17: Reconstruction using model trained with $\lambda = 0.01$

(a) MS-SSIM rate-distortion curve for Kodak image 18.

(b) PSNR rate-distortion curve for Kodak image 18.

(c) Kodak image 18: Original

(d) Kodak image 18: Reconstruction using model trained with $\lambda = 0.01$

(a) MS-SSIM rate-distortion curve for Kodak image 19.

(b) PSNR rate-distortion curve for Kodak image 19.

(c) Kodak image 19: Original

(d) Kodak image 19: Reconstruction using model trained with $\lambda = 0.01$

(a) MS-SSIM rate-distortion curve for Kodak image 20.

(b) PSNR rate-distortion curve for Kodak image 20.

(c) Kodak image 20: Original

(d) Kodak image 20: Reconstruction using model trained with $\lambda = 0.01$

(a) MS-SSIM rate-distortion curve for Kodak image 21.

(b) PSNR rate-distortion curve for Kodak image 21.

(c) Kodak image 21: Original

(d) Kodak image 21: Reconstruction using model trained with $\lambda = 0.01$

(a) MS-SSIM rate-distortion curve for Kodak image 22.

(b) PSNR rate-distortion curve for Kodak image 22.

(c) Kodak image 22: Original

(d) Kodak image 22: Reconstruction using model trained with $\lambda = 0.01$

(a) MS-SSIM rate-distortion curve for Kodak image 23.

(b) PSNR rate-distortion curve for Kodak image 23.

(c) Kodak image 23: Original

(d) Kodak image 23: Reconstruction using model trained with $\lambda = 0.01$

(a) MS-SSIM rate-distortion curve for Kodak image 24.

(b) PSNR rate-distortion curve for Kodak image 24.

(c) Kodak image 24: Original

(d) Kodak image 24: Reconstruction using model trained with $\lambda = 0.01$

## Footnotes

[9] For Gaussian random variables $X, Y$ with means $\mu_x, \mu_y$ and variances $\sigma_x^2, \sigma_y^2$ and $Z = X + Y$, $p(x|z)$ is normally distributed with mean $\mu_x + (z - \mu_x - \mu_y)\frac{\sigma_x^2}{\sigma_x^2 + \sigma_y^2}$ and variance $\frac{\sigma_x^2 \sigma_y^2}{\sigma_x^2 + \sigma_y^2}$