[Reviews · NeurIPS 2020]

Review 1

Summary and Contributions: This paper aims to resolve two limitations of bits-back compression: (a) transmission of the auxiliary bitstream degrades the rates, (b) bits-back coding only applies to the lossless compression. The authors propose an index coding (iREC) which allows encoding the latent representation z with codelength close to the relative entropy. The key idea is to use importance sampling and index coding together with shared randomness, where p(z) is assumed to be shared and multiple realizations of z ~ p(z) can be shared between the encoder and decoder. For computational reasons, the authors propose using auxiliary variables to represent z and apply important sampling on the auxiliary space.

Strengths: I think the biggest contribution is in the algorithm itself. The proposed idea of index coding together with importance sampling, introduction of auxiliary variables and beam search, is interesting and seems novel. The problem considered is a single image compression, which may be a narrow application, but I think it has potential usage. For the single image compression setting, the empirical results show iREC outperforms other the variants of bits-back compression.

Weaknesses: The authors articulate several benefits of iREC, but those were not fully convincing. i) Lossy compression: While the proposed method is immediately applicable to lossy compression, the performance on lossy compression seems to be clearly worse than several other baselines. ii) Being able to handle continuous latent space is a plus, but the gain from that ability was not clear. Another limitation is, as mentioned by the authors, its computation cost.

Correctness: I have a question regarding the statement "We present Index coding (iREC), a REC algorithm that scales to ...." (line 163) Does iREC provably achieve communication cost provably close to KLD? The authors state "We refer to algorithms that achieve communication cost provably close to KL [ q(z) || p(z) ] as REC algorithms." (line 151)

Clarity: The paper is overall well written. The author provides a high-level intuition at various places (e.g., stochastic coding scheme), which I think greatly helps readers to follow the paper.

Relation to Prior Work: The relation to prior and directly related work was presented clearly.

Reproducibility: Yes

Additional Feedback:


Review 2

Summary and Contributions: This paper propose a method with relative entropy coding that can be applied both lossless and lossy image compression. The latent representation is coded with codelength close to the relative entropy for single images, which can outperform existing bits-back methods that requires auxiliary bits.

Strengths: This paper contains complete theoretical derivation, with very detailed algorithm description with source code in supplementary materials. It provides a new lossless image compression pipeline based on VAE. The proposed method shows much bitrate save when comparing with existing bits-back methods on single image.

Weaknesses: The performance compared with no bits-back method is not that competitive. The highest resolution image used in the paper is Kodak, but lossless single image lossless compression might be more essential for images with higher resolution. Besides, the proposed method show clear performance gap with state-of-the-art methods in lossy image compression, but its provides an new perspective for lossy image compression.

Correctness: Yes, except that in Tab.1 the proposed method should not be included in the bits-back methods, since it is fundamentally different from those bits-back methods.

Clarity: Yes

Relation to Prior Work: Yes

Reproducibility: Yes

Additional Feedback:


Review 3

Summary and Contributions: The paper presents index coding based REC method (iREC) to extend the use of VAE (continuous latent variable) for both lossless and lossy compression. The proposed method can encode an image with codelength close to the negative ELBO. The paper presents experiments on benchmark datasets and demonstrate improved performance on both lossless and lossy compression task.

Strengths: - The presented algorithm (iREC) is a novel approach (in terms of a coding scheme) that proposes the sampling of auxiliary variables instead of directly sampling from the learned posterior of the latent variables. - The presented results are impressive as the method is outperforming others recently proposed method in the compression literature in case of lossless compression and in the case of lossy compression, the paper presents comparable performance to the current state-of-the-art method.

Weaknesses: - One of the biggest weaknesses is the clear discussion/motivation on why other methods are problematic. This is important because the use of continuous latent space in this kind of problem has been rigorously discussed in the ML literature and without clearly differentiating with prior works, the contribution is weakened. - The experiments for ablation is missing. For example, the effect of bias of importance sampling (or benefit of the beam search) is not demonstrated empirically.

Correctness: Some of the claims are not validated empirically (missing ablation studies). Empirical methodology correct.

Clarity: The paper is poorly written with a lot of typos, and unclear and vague statements. E.g., - "who to our knowledge" : line 255 - "As our model we used the ..": line 257

Relation to Prior Work: The relation to prior work is made vague. The statements like "We introduce REC" in line 47 contradicts with "they present a REC algorithm" (line 160).

Reproducibility: Yes

Additional Feedback: - Authors are encouraged to clearly validate all the claims made in the paper empirically. - The proofreading and correcting grammatical errors could enhance the strength of the paper.


Review 4

Summary and Contributions: This paper proposes a method to compress the latent representation generated by the encoder of an autoencoder. The method is abased on a bits-back scheme and addresses the problem of such scheme requiring a large context to be effective, ie being inefficient to encode a single image.

Strengths: The addressed problem, ie compression of the latent space variables of an autoencoder is definitely relevant and timely. The paper is extremely well written and contains a good tutorial on bits-back methods that make the paper accessible.

Weaknesses: 32x32 Cifar and Imagenet are a bit unsuited to evaluate the quality of the proposed scheme. The Kodak dataset is much more appropriate, however it starts to be a bit outdated nowadays. The authors could for example consider the Class B test sequences of MPEG for HEVC or VVC. I would also like to see JPEG2000 in Fig 3a) and 3b)

Correctness: Yes.

Clarity: Very well written, good tutorial.

Relation to Prior Work: Yes.

Reproducibility: Yes

Additional Feedback:

[Author Response · NeurIPS 2020]

**Summary**   We thank the reviewers for providing important feedback to improve our work. We are delighted that the reviewers all recognized the relevance and novelty of the core contribution of our paper, the iREC compression algorithm (R1, R2, R3, R4), and noted its benefits in terms of compression performance both in the lossless (R1, R2, R3) and in the lossy domain (R3). We are also pleased that the reviewers found our paper to be clearly written (R1, R2, R4), our methodology correct (R2, R3, R4) and all judged our results to be reproducible. The reviewers had the following main concerns: **Performance:** (R1, R2) were concerned about the performance of our method against the state-of-the-art. **Datasets Used:** (R2, R4) voiced concerns with the datasets used in our experiments. **Motivation:** R3 asked us to expand on the motivation for our work and noted the lack of an **Ablation Study** to show the benefits of the beam search procedure discussed in Section 3.1.2. We address all four concerns below.

**Performance**   We want to highlight that both Figure 3a and 3b show the performance on a logarithmic scale, which has the effect of over-emphasizing performance at high bitrates and under-emphasizing performance at low bitrates. Figure 1 shows the same data represented on a linear scale. We can see that iREC and Ballé (2018) outperform the competing baseline methods and that the difference between Ballé (2018) and iREC is imperceptible at all bitrates.

Figure 1: MS-SSIM rate-distortion curves

**Datasets Used**   We agree with the reviewers that the CIFAR-10 and ImageNet32 datasets are inadequate to assess true performance well, hence the inclusion of the Kodak dataset in Table 1. We included the low-resolution comparisons to compare against competing bits-back methods that are yet to be scaled to high-resolution images. For the lossy compression comparisons, we opted to use the Kodak dataset due to its ubiquity as a benchmark test dataset for ML-based image compression methods. We will consider the Class B test sequences (suggested by R4) in our future works.

**Motivation**   The motivation to our work is to bring the benefits of bits-back coding to the compression of single images and to the lossy compression domain. The first benefit is the ability to work with continuous latent distributions, which greatly expands the viable generative models for compression and simplifies training, and the second benefit is the theoretical guarantee on the codelength.

We agree with R3 that in the context of generative modelling, the use of continuous latent distributions has been studied extensively. In Section 2.1 we discuss the related approaches in the compression setting: we describe and contrast quantization approaches that use discrete latent distributions, and BB-ANS that uses continuous latent distributions. We are keen to expand on this discussion, please let us know if we missed any prominent works.

**Ablation Study**   We thank R3 for pointing out the lack of mention of an ablation study in the main text. We performed several ablation studies, the results of which are reported in Figures 1 and 2 in the supplementary material, where we experimented with several settings for $\Omega$, $\epsilon$ and $B$. In the special case of a single search beam ($B = 1$), the algorithm is equivalent to the importance sampling procedure described in Section 3.1.1. The most relevant details are in rows 1 and 3 of Figure 1, where both the coding overhead (row 1) and the residual overhead (row 3, a proxy for the bias of the sampling procedure) are shown to be much worse compared to using the beam search procedure, across a variety of different settings. In the final version of the paper we will include a clear discussion about these studies.

**Further Questions and Issues**

*Provably close communication cost of iREC to the KL* (R1): Since we draw $S \approx \exp(\text{KL})$ samples, the index of a sample can be communicated in approx. $\log S = \text{KL}$ nats (lines 170 - 175). We refer to Havasi et al. (2019) for the rigorous proof of correctness for the importance sampling procedure.

*Gain from using continuous latent spaces* (R1): We agree with the reviewer that we do not demonstrate significant performance benefits from this in the paper, and investigating how to best leverage this is left for future work.

*Separating our method from bits-back methods in Tab. 1* (R2): We agree and we will separate them in the final version.

*Typos and grammatical errors* (R3): We thank the reviewer for bringing these to our attention, and we will correct these errors in the final version of the paper. Regarding line 160, we agree that it is confusing and potentially misleading and will clarify this point in the final version of the paper.

*Including JPEG2000 as a benchmark in Figures 3a and 3b* (R4): We opted against including more benchmarks in the figures to avoid clutter. We will make sure to include a more detailed figure in the supplementary material.

**Thank you for reviewing our work. If this response adequately addressed your concerns, please consider adjusting your score.**

[Meta-Review · NeurIPS 2020]

The approach is novel, presented in a nice and clear way and with good results. After the rebuttal and discussion phase, there was a generalized consensus that this submission should be accepted to the conference. That said, a reviewer raised concerns regarding an ablation experiment on the effect of importance sampling. I would encourage the authors to consider, space permitting, such an addition, or at least to add a sentence to clarify this. Some typos are also present and should be fixed.